# Deep Dependency Networks for Action Classification in Videos

## Abstract

We propose a simple approach which combines the strengths of probabilistic graphical models and deep learning architectures for solving the multi-label action classification task in videos. At a high level, given a video clip, the goal in this task is to infer the set of activities, defined as verb-noun pairs, that are performed in the clip. First, we show that the performance of previous approaches that combine Markov Random Fields with neural networks can be modestly improved by leveraging more powerful methods such as iterative join graph propagation, $\ell_1$ regularization based structure learning and integer linear programming. Then we propose a new modeling framework called deep dependency network which augments a dependency network, a model that is easy to train and learns more accurate dependencies but is limited to Gibbs sampling for inference, to the output layer of a neural network. We show that despite its simplicity, joint learning this new architecture yields significant improvements in performance over the baseline neural network. In particular, our experimental evaluation on three video datasets: Charades, Textually Annotated Cooking Scenes (TACoS), and Wetlab shows that deep dependency networks are almost always superior to pure neural architectures that do not use dependency networks.

## 1 Introduction

We focus on the following multi-label action classification (MLAC) task: given a video partitioned into segments (or frames) and a pre-defined set of actions, label each segment with a subset of actions from the pre-defined set that are performed in the segment (or frame). We consider each action to be a verb-noun pair such as "open bottle", "pour water" and "cut vegetable". MLAC is a special case of the standard multi-label classification (MLC) task where given a pre-defined set of labels and a test example, the goal is to assign each test example to a subset of labels.

It is well known that MLC is notoriously difficult because in practice the labels are often correlated and thus predicting them independently may lead to significant errors. Therefore, most advanced methods explicitly model the relationship or dependencies between the labels, either using probabilistic techniques (cf. Wang et al. (2008); Guo & Xue (2013); Tan et al. (2015); Di Mauro et al. (2016); Wang et al. (2014); Antonucci et al. (2013)) or non-probabilistic/neural methods (cf. Papagiannopoulou et al. (2015); Kong et al. (2013); Wang et al. (2021a); Nguyen et al. (2021); Wang et al. (2021b)). Intuitively, because MLAC is a special case of MLC, methods used for solving MLAC should model relationships between the actions as well as between the actions and features. To this end, the primary goal of this paper is to develop a simple, general-purpose scheme that models the relationships between actions using probabilistic representation and reasoning techniques and works on top of neural feature extractors in order to improve their generalization performance.

Our work is inspired by previous work on hybrid models that combine probabilistic graphical models (PGMs) with neural networks (NNs) under the assumption that their strengths are often complementary (cf. Johnson et al. (2016); Krishnan et al. (2015). At a high level, in these hybrid models NNs perform feature extractions while PGMs model the relationships between the labels as well as between features and labels. In previous work, these hybrid models have been used for solving a range of computer vision tasks such as image crowd counting (Han et al., 2017), visual relationship detection (Yu et al., 2022), modeling for epileptic seizure detection in multichannel EEG (Craley et al., 2019), face sketch synthesis (Zhang et al., 2020), semantic image segmentation (Chen et al.,

2018; Lin et al., 2016), animal pose tracking (Wu et al., 2020) and pose estimation (Chen & Yuille, 2014). In this paper, we seek to extend and adapt these previously proposed PGM+NN approaches for solving the multi-label action classification task in videos.

Motivated by approaches that combine PGMs with NNs, as a starting point, we investigated using Markov random fields (MRFs), a type of undirected PGM to capture the relationship between the labels and features computed using convolutional NNs. Unlike previous work which used these MRF+CNN or CRF+CNN hybrids with conventional inference schemes such as Gibbs sampling (GS) and mean-field inference, our goal was to evaluate whether advanced reasoning and learning approaches, specifically (1) iterative join graph propagation (IJGP) (Mateescu et al., 2010) (a type of generalization Belief propagation technique Yedidia et al. (2000)), (2) integer linear programming (ILP) based techniques for computing most probable explanations, and (3) logistic regression with $\ell_1$-regularization based methods (Lee et al., 2006; Wainwright et al., 2006) for learning the structure of pairwise MRFs, improve the generalization performance of MRF+CNN hybrids. To measure generalization accuracy, we used several metrics such as mean average precision (mAP), label ranking average precision (LRAP), subset accuracy (SA) and the jaccard index (JI) and experimented on three video datasets: (1) Charades (Sigurdsson et al., 2016), (2) TACoS (Regneri et al., 2013) and (3) Wetlab (Naim et al., 2014). We found that, generally speaking, both IJGP and ILP are superior to the baseline CNN and Gibbs sampling in terms of JI and SA but are sometimes inferior to the CNN in terms of mAP and LRAP. We speculated that because MRF structure learners only allow pairwise relationships and impose sparsity or low-treewidth constraints for faster, accurate inference, they often yield poor posterior probability estimates in high-dimensional settings. Since both mAP and LRAP require good posterior probability estimates, GS, IJGP, and ILP exhibit poor performance when mAP and LRAP are used to evaluate the performance.

To circumvent this issue and in particular, to derive good posterior estimates, we propose a new PGM+NN hybrid called *deep dependency networks* (DDNs). At a high level, a dependency network (DN) (Heckerman et al., 2000) represents a joint probability distribution using a collection of conditional distributions, each defined over a variable given all other variables in the network. A DN is consistent if the conditional distributions come from a unique joint probability distribution. Otherwise, it is called inconsistent. A consistent DN has the same representation power as a MRF in that any MRF can be converted to a consistent DN and vice versa. A key advantage of DNs over MRFs is that they are easy to train because each conditional distribution can be trained independently and modeled using classifiers such as logistic regression, decision trees, and multi-layer perceptrons. These classifiers can be easily defined over a large number of features and as a result the conditional distributions (defined by the classifiers) are often more accurate than the ones inferred from a sparse or low-treewidth MRF learned from data. However, DNs admit very few inference schemes and are more or less restricted to Gibbs sampling for inference in practice. The second disadvantage of DNs is that because the local distributions are learned independently, they can yield an inconsistent DN, and thus one has to be careful when performing inference over such DNs. Despite these disadvantages, DNs often learn better models which typically translates to superior generalization performance (cf. (Neville & Jensen, 2003; Guo & Gu, 2011)).

In our proposed deep dependency network (DDN) architecture, a dependency network sits on the top of a convolutional neural network (CNN). The CNN converts the input image or video segment to a set of features, and the dependency network uses these features to define a conditional distribution over each label (action) given the features and other labels. We show that deep dependency models are easy to train either jointly or via a pipe-line method where the CNN is trained first, followed by the DNN by defining an appropriate loss function that minimizes the negative pseudo log-likelihood of the data. We conjecture that because DDNs can be quite dense, they often learn a better representation of the data, and as a result, they are likely to outperform MRFs learned from data in terms of posterior predictions.

We evaluated DDNs using the four aforementioned metrics and three datasets. We observed that they are often superior to the baseline neural networks as well as MRF+CNN methods that use GS, IJGP and ILP on all four metrics. Specifically, they achieve the highest score on the mAP metric on all the datasets. We compared the pipeline model with the jointly learned model and found that the joint model is more accurate than the pipeline model.

In summary, this paper makes the following contributions:

- We propose a new hybrid model called deep dependency networks that combines the strengths of dependency networks (faster training and ability to use advanced classifiers) and neural networks (flexibility, feature representation).

- We define a new loss function for jointly training the neural and dependency network components of the deep dependency network. We show that DDNs are easy to train using these loss functions.

- We experimentally evaluate DDNs on three video datasets and using four metrics for solving the multi-label action classification task. We found that jointly trained DDNs consistently outperform the baseline CNNs and MRF+CNN hybrids on all metrics and datasets.

## 2 PRELIMINARIES

A *log-linear model* or a *Markov random field* (MRF), denoted by $\mathcal{M}$, is an undirected probabilistic graphical model (cf. Koller & Friedman (2009)) that is widely used in many real-world domains for representing and reasoning about uncertainty. It is defined as a triple $\langle \mathbf{X}, \mathcal{F}, \Theta \rangle$ where $\mathbf{X} = \{X_1, \ldots, X_n\}$ is a set of Boolean random variables, $\mathcal{F} = \{f_1, \ldots, f_m\}$ is a set of features such that each feature $f_i$ (we assume that a feature is a Boolean formula) is defined over a subset $\mathbf{D}_i$ of $\mathbf{X}$, and $\Theta = \{\theta_1, \ldots, \theta_m\}$ is a set of real-valued weights or parameters, namely $\forall \theta_i \in \Theta; \ \theta_i \in \mathbb{R}$ such that each feature $f_i$ is associated with a parameter $\theta_i$. $\mathcal{M}$ represents the following probability distribution:

$$P(\mathbf{x}) = \frac{1}{Z(\Theta)} \exp \left\{ \sum_{i=1}^{k} \theta_i f_i \left( \mathbf{x}_{\mathbf{D}_i} \right) \right\} \tag{1}$$

where $\mathbf{x}_{\mathbf{D}_i}$ is the projection of $\mathbf{x}$ on the variables $\mathbf{D}_i$ of the feature $f_i$, $f_i(\mathbf{x}_{\mathbf{D}_i})$ is an *indicator function* that equals 1 when the assignment $\mathbf{x}_{\mathbf{D}_i}$ evaluates $f_i$ to `True` and is 0 otherwise, and $Z(\Theta)$ is the normalization constant called the *partition function*.

We focus on three tasks over MRFs: (1) structure learning which is the problem of learning the features and parameters given training data; (2) posterior marginal inference which is the task of computing the marginal probability distribution over each variable in the network given evidence (evidence is defined as an assignment of values to a subset of variables); and (3) finding the most likely assignment to all the non-evidence variables given evidence (this task is often called maximum-a-posteriori or MAP inference in short). All of these tasks are at least NP-hard in general and therefore approximate methods are often preferred over exact ones in practice.

A popular and fast method for structure learning is to learn binary pairwise MRFs by training an $\ell_1$-regularized logistic regression classifier for each variable given all other variables as features (Wainwright et al., 2006; Lee et al., 2006). $\ell_1$-regularization induces sparsity in that it encourages many weights to take the value zero. All non-zero weights are then converted into conjunctive features. Each conjunctive feature evaluates to `True` if both variables are assigned the value 1 and to `False` otherwise. Popular approaches for posterior marginal inference are the Gibbs sampling algorithm and generalized Belief propagation (Yedidia et al., 2000) techniques such as Iterative Join Graph Propagation (Mateescu et al., 2010). For MAP inference, a popular approach is to encode the optimization problem as a linear integer programming problem (cf. Koller & Friedman (2009)) and then using off-the-shelf approaches such as Gurobi Optimization, LLC (2022) to solve the latter.

**Dependency Networks (DNs)** (Heckerman et al., 2000) represent the joint distribution using a set of conditional probability distributions, one for each variable. Each conditional distribution defines the probability of a variable given all of the others. A DN is consistent if there exists a joint probability distribution $P(\mathbf{x})$ such that all conditional distributions $P_i(x_i | \mathbf{x}_{-i})$ where $\mathbf{x}_{-i}$ is the projection of $\mathbf{x}$ on $\mathbf{X} \setminus \{X_i\}$, are conditional distributions of $P(\mathbf{x})$.

A DN is learned from data by learning a probabilistic classifier (e.g., logistic regression, multi-layer perceptron, etc.) for each variable, and thus DN learning is embarrassingly parallel. However, because the classifiers are independently learned from data, we often get an inconsistent DN. It has been conjectured (Heckerman et al., 2000) that most DNs learned from data are almost consistent in that only a few parameters need to be changed in order to make them consistent.

The most popular inference method over DNs is *fixed-order* Gibbs sampling (Liu, 2008). If the DN is consistent, then its conditional distributions are derived from a joint distribution $P(\mathbf{x})$, and

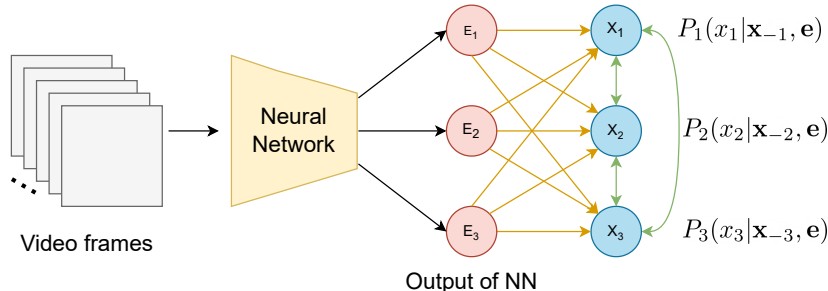

Figure 1: Dependency Network structure for three actions. Video frames are given as input to the NN and it produces the features for the DN $(E_1, E_2, E_3)$. These features are then used to model the conditional distributions at each variable $X_i$. At each node $X_i$, the form of the conditional distribution is the variable given its parents (incoming arrows) in the graph.

the stationary distribution (namely the distribution that Gibbs sampling converges to) will be the same as $P(\mathbf{x})$. If the DN is inconsistent, then the stationary distribution of Gibbs sampling will be inconsistent with the conditional distributions.

## 3 DEEP DEPENDENCY NETWORKS

In this section, we describe how to solve the multi-label action classification task in videos using a hybrid of dependency networks and neural networks. At a high level, the neural network provides high quality features given videos and the dependency network represents and reasons about the probabilistic relationships between the labels and features.

### 3.1 THE MODEL

Let $\mathbf{V}$ denote the set of random variables corresponding to the pixels and $\mathbf{v}$ denote the RGB values of the pixels in a frame or a video segment. Let $\mathbf{E}$ denote the (continuous) output nodes of a neural network which represents a function $\mathbb{N} : \mathbf{v} \to \mathbf{e}$, that takes $\mathbf{v}$ as input and outputs an assignment $\mathbf{e}$ to $\mathbf{E}$. Let $\mathbf{X} = \{X_1, \ldots, X_n\}$ denote the set of actions (labels). For simplicity, we assume that $|\mathbf{E}| = |\mathbf{X}| = n$. Given $(\mathbf{V}, \mathbf{E}, \mathbf{X})$, a deep dependency network (DDN) is a pair $\langle \mathcal{N}, \mathcal{D} \rangle$ where $\mathcal{N}$ is a neural network that maps $\mathbf{V} = \mathbf{v}$ to $\mathbf{E} = \mathbf{e}$ and $\mathcal{D}$ is a conditional dependency network (Guo & Gu, 2011) that models $P(\mathbf{x}|\mathbf{e})$ where $\mathbf{e} = \mathbb{N}(\mathbf{v})$. The conditional dependency network represents the distribution $P(\mathbf{x}|\mathbf{e})$ using a collection of local conditional distributions $P_i(x_i|\mathbf{x}_{-i}, \mathbf{e})$, one for each action (label) $X_i$, where $\mathbf{x}_{-i} = \{x_1, \ldots, x_{i-1}, x_{i+1}, \ldots, x_n\}$. Fig. 1 shows the network architecture for $n = 3$. Note that the goal is to model $P(\mathbf{x}|\mathbf{v})$ as accurately as possible and we assume that $P(\mathbf{x}|\mathbf{v}) = P(\mathbf{x}|\mathbf{e})$. Our hope is that $P(\mathbf{x}|\mathbf{e})$ is easier to learn and represent as compared to $P(\mathbf{x}|\mathbf{v})$, and thus the loss functions at the neural network and the conditional dependency network must be carefully chosen to facilitate this objective.

### 3.2 INFERENCE

Once a DDN is learned from data, at test time, it can be used to provide labels for a given video frame or segment as follows. We first send the video frame $\mathbf{v}$ through the neural network $\mathcal{N}$ to yield an assignment $\mathbf{e}$ to the output nodes of the neural network. Then, we perform fixed-order Gibbs Sampling over the conditional dependency network where the latter represents the distribution $P(\mathbf{x}|\mathbf{e})$. Finally, given samples $(\mathbf{x}^{(1)}, \ldots, \mathbf{x}^{(N)})$ generated via Gibbs sampling, we estimate the marginal probability distribution of each label $X_i$ using the following mixture estimator (Liu, 2008)

$$\hat{P}(x_i|\mathbf{v}) = \frac{1}{N} \sum_{j=1}^{N} P_i \left( x_i^{(j)} \mid \mathbf{x}_{-i}^{(j)}, \mathbf{e} \right).$$

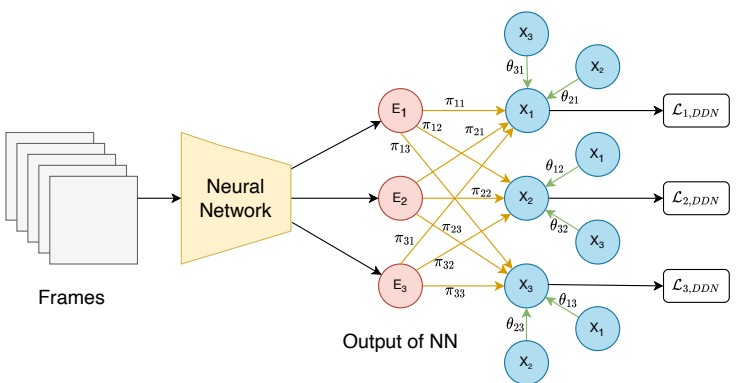

Figure 2: Computational graph for the forward pass. Each edge is associated with a weight (parameter), either $\theta_{ij}$ or $\pi_{ij}$.

### 3.3 LEARNING

We can either train the DDN using a pipeline method or via joint training. In the pipeline method, we first train the neural network using conventional approaches (e.g., using cross-entropy loss) or we can use a pre-trained model. Then for each training example $(\mathbf{v}, \mathbf{x})$, we send the video through the neural network to obtain a new representation $\mathbf{e}$ of $\mathbf{v}$. The aforementioned process transforms each training example $(\mathbf{v}, \mathbf{x})$ into a new feature representation $(\mathbf{e}, \mathbf{x})$ where $\mathbf{e} = \mathbb{N}(\mathbf{v})$. Finally, for each action (label) $X_i$, we learn a probabilistic classifier to model the conditional distribution $P_i(x_i|\mathbf{x}_{-i}, \mathbf{e})$ using the transformed training data. Specifically, given a training example $(\mathbf{e}, \mathbf{x})$, each probabilistic classifier indexed by $i$, uses $X_i$ as the class variable and $(\mathbf{E} \cup \mathbf{X}_{-i})$ as the attributes. In our experiments, we used two classifiers, logistic regression and multi-layer perceptron.

A key advantage of using the pipeline method is that training is relatively fast and can be easily parallelized. Moreover, it requires only modest computational resources. As a result, the pipeline method is especially beneficial in the following scenario: limited computational resources are available but a pre-trained neural network that is trained on high-powered GPUs is readily available.

In practice, in order to perform joint training, we need to select appropriate loss functions defined over the parameters of the conditional distributions and then perform backpropagation/gradient descent to minimize the loss functions. Intuitively, given a pre-trained model with output nodes $\mathbf{E}$ and a set of probabilistic classifiers which model $P(X_i|\mathbf{e}, \mathbf{x}_{-i})$, we can simply define an appropriate loss function at $X_i$ and back-propagate it to the neural network. For instance, given a training example $(\mathbf{v}, \mathbf{x})$ and under the assumption that a logistic regression classifier with parameters $(\Theta_i, \Pi_i)$ is used to model $P_i(X_i|\mathbf{e}, \mathbf{x}_{-i})$ where $\Theta_i$ are weights associated with $\mathbf{x}_{-i}$ and $\Pi_i$ are weights associated with $\mathbf{e}$, we can use the following cross-entropy loss at each classifier indexed by $i$:

$$\mathcal{L}_{i,DDN} = -x_i \log P_i(X_i = 1|\mathbf{e}, \mathbf{x}_{-i}; \Theta_i, \Pi_i) - (1 - x_i) \log P_i(X_i = 0|\mathbf{e}, \mathbf{x}_{-i}; \Theta_i, \Pi_i) \quad (2)$$

We can then back-propagate this loss via the parameters $\Pi_i$ to each output node of the NN.

Fig. 2 shows an example computation graph (forward pass) associated with the loss function given in equation 2 for three labels. At a high level, when logistic regression is used, at learning time, the loss function alters the structure of the neural network by adding an extra layer (dependency network) as shown. While training, this layer acts as multiple MLPs (each representing the distribution for $X_i$). However, note that at test time, we have to use Gibbs sampling as described in the previous section.

An issue with the loss function just described is that large parameter gradients from the dependency network may rapidly change the parameters of the pre-trained network; thus undoing much of the work done during pre-training. Therefore, we propose to use the following loss function at each output node of the neural network

$$\mathcal{L}_{i,NN} = (-x_i \log e_i - (1 - x_i) \log(1 - e_i)) + \frac{\lambda}{n} \sum_{j=1}^{n} \mathcal{L}_{j,DDN} \quad (3)$$

where $\lambda \geq 0$ is a hyper-parameter. Note that the loss given in equation 3 is a generalization of the one given in equation 2 because for large values of $\lambda$, the effect of the first two terms disappears.

The main advantage of the loss given in equation 3 is that it allows us to dampen the negative effects of any wrong predictions made at the dependency network layer, on the correctly predicted output node in the neural network. For example, in Fig. 2, let us consider node $E_1$ which is connected to all nodes in the dependency network. Let us assume that $E_1$ and $X_1$ were correctly predicted for a training example, while the predictions at $X_2$ and $X_3$ were incorrect. In such situations, we do not want $E_1$ to be unnecessarily penalized (since it was correctly predicted). However, the loss given in equation 2 will penalize $E_1$ via backpropagation through the edges $(E_1, X_2)$ and $(E_1, X_3)$. On the other hand, the loss given in equation 3 will dampen the effect via $\lambda$.

Our proposed loss function (see equation 3) is similar to the ones used in multi-task learning (cf. (Caruana, 1997)) where we have two or more networks that share a substructure. In our setting, we have two networks, one which explicitly reasons about dependencies between the labels and another which does not.

## 4 EXPERIMENTAL EVALUATION

In this section, we evaluate the proposed models on the multi-label activity classification task using three video datasets. We begin by describing the datasets and metrics, followed by the experimental setup and conclude with the results. All models were implemented using PyTorch, and one NVIDIA A40 GPU was used to train and test all the models.

### 4.1 DATASETS AND METRICS

We evaluated our algorithms on the following three video datasets: (1) Charades (Sigurdsson et al., 2016); (2) Textually Annotated Cooking Scenes (TACoS) (Regneri et al., 2013); and (3) Wetlab (Naim et al., 2015). In the Charades dataset, the videos are divided into segments (video clips) and each segment is annotated with one or more action labels. In the TACoS and Wetlab datasets, each frame is associated with one or more actions.

**Charades Dataset** (Sigurdsson et al., 2016) comprises of videos of people performing daily indoor activities while interacting with various objects. It is a multi-label activity classification dataset; thus, more than one activity can happen at a given time. In the standard split, there are 7,986 training videos and 1,863 validation videos, averaging 30 seconds. We used the training videos to train the models and the validation videos for testing purposes (because the authors have not released the test set). The dataset has roughly 66,500 temporal annotations for 157 action classes. We report mean Average Precision (mAP) following the standard evaluation protocols for this dataset. We also report Label Ranking Average Precision (LRAP) and Jaccard Index (JI), as these metrics have been used extensively in previous work for evaluating the performance of multi-label classifiers.

The **Textually Annotated Cooking Scenes (TaCOS)** dataset (Regneri et al., 2013) consists of third-person videos of a person cooking in a kitchen. The dataset comes with hand-annotated labels of actions, objects, and locations for each video frame. From the complete set of these labels, we selected 28 labels to create our dataset. Each of these labels corresponds to a location, an object, or a verb, and each action is defined as a (location, object, verb) triple. By selecting the videos that correspond to these labels and dividing them into train and test sets, we get a total of 60313 frames for training and 9355 frames for testing, spread out over 17 videos.

The **Wetlab dataset** (Naim et al., 2014) is comprised of videos where experiments are being performed by lab technicians that involve hazardous chemicals in a wet laboratory. The dataset contains 6 videos where each video follows three protocols: Cellobiose M9 Media (CELL), LB Liquid Growth Media (LLGM), and Yeast (YPAD) Media. We used one video for testing and five videos for training. Specifically, the training set comprises of 100054 frames, and the test set comprises of 11743 frames. Each activity in the dataset can have multiple objects per activity (for example, an activity "transfer contents" can have two objects linked with it, a test tube and a beaker) and there are 57 possible labels. Each label corresponds to an object or a verb, and each action is made of one or more labels from each category.

We evaluated the performance on the TACoS and Wetlab datasets using the following four metrics: mean Average Precision (mAP), Label Ranking Average Precision (LRAP), Subset Accuracy (SA), and Jaccard Index (JI). Note that SA is not relevant for the Charades dataset. mAP and LRAP require access to an estimate of the posterior probability distribution at each label while SA and JI are non-probabilistic and only require an assignment to the labels.

## 4.2 Experimental Setup and Methods

We used three types of architectures in our experiments: (1) Baseline convolutional neural networks (CNNs) which are specific to each dataset; (2) CNNs augmented with MRFs, which we will refer to as deep random fields or DRFs in short; and (3) The method proposed in this paper which uses a dependency network on top of a CNN called deep dependency networks (DDNs).

**Convolutional Neural Networks**. We choose two different types of models for CNNs, and they act as a baseline for the experiments and as a feature extractor for DRFs and DDNs. We chose two types of models for this, 2D CNNs and 3D CNNs. In doing this, we want to demonstrate that the proposed method can work well for two different types of feature extractors. For the Charades dataset we use the PySlowFast (Fan et al., 2020) implementation of the SlowFast Network (Feichtenhofer et al., 2019) which uses a 3D ResNet model as the backbone. For the TACoS and Wetlab datasets, we use InceptionV3 (Szegedy et al., 2016), one of the state-of-the-art 2D CNN models for image classification. For extracting the features from the Charades dataset, we use the pre-trained model and hyper-parameters given in PySlowFast. For the TaCOS and Wetlab datasets, we used a InceptionV3 model pre-trained on the ImageNet data set and then retrained it for each dataset.

**Deep Random Fields (DRFs)**. As a baseline, we used a model that combines MRFs with CNNs. This DRF model is similar to the DDN except that we use a MRF instead of a DDN to compute $P(\mathbf{x}|\mathbf{e})$. We trained the MRFs generatively, namely we learned a joint distribution $P(\mathbf{x}, \mathbf{e})$, which can be used to compute $P(\mathbf{x}|\mathbf{e})$ by instantiating evidence. We chose generative learning because we learned the structure of the MRFs from data and discriminative structure learning is slow in practice (cf. Koller & Friedman (2009)). Specifically, we used the logistic regression with $\ell_1$ regularization method of Wainwright et al. (2006) to learn a pairwise MRF. The training data for this method is obtained by sending each annotated video clip (or frame) $(\mathbf{v}, \mathbf{x})$ through the CNN and transforming it to $(\mathbf{e}, \mathbf{x})$ where $\mathbf{e} = \mathbb{N}(\mathbf{v})$. At termination, this method yields a graph $\mathcal{G}$ defined over $\mathbf{X} \cup \mathbf{E}$.

For parameter/weight learning, we converted each edge over $\mathbf{X} \cup \mathbf{E}$ to a conjunctive feature. For example, if the method learns an edge between $X_i$ and $E_j$, we use a conjunctive feature $X_i \wedge E_j$ which is true iff both $X_i$ and $E_j$ are assigned the value 1. Then we learned the weights for each conjunctive feature by maximizing the pseudo log-likelihood of the data (Besag, 1975).

For inference over MRFs, we used Gibbs sampling (GS), Iterative Join Graph Propagation (IJGP) Mateescu et al. (2010) and Integer Linear Programming (ILP) methods. Thus, we used three versions of DRFs corresponding to the inference scheme used. We refer to these schemes as DRF-GS, DRF-IJGP and DRF-ILP respectively. Note that both IJGP and ILP are advanced schemes, and we are not aware of their use for multi-label action classification. Our goal is to test whether advanced inference schemes help improve the performance of deep random fields.

**Deep Dependency Networks (DDNs)**. We experimented with four versions of DDNs: (1) DDN-LR-Pipeline; (2) DDN-MLP-Pipeline; (3) DDN-LR-Joint; and (4) DDN-MLP-Joint. The first and the third versions use logistic regression (LR) while the second and fourth versions use multi-layer perceptrons (MLP) respectively to represent the conditional distributions. The first two versions are trained using the pipeline method while the last two versions are trained using the joint learning loss given in equation 3.

**Hyperparameters:** For DRFs, in order to learn a sparse structure (using the logistic regression with $\ell_1$ regularization method of Wainwright et al. (2006)), we increased the regularization constant associated with the $\ell_1$ regularization term until the number of neighbors of each node in $\mathcal{G}$ is bounded between 2 and 10. We enforced this sparsity constraint in order to ensure that the inference schemes (specifically, IJGP and ILP) are accurate, and the model does not overfit the training data. IJGP, ILP and GS are anytime methods and for each, we used a time-bound of 60 seconds per example.

For DDNs, we used LR with $\ell_2$ regularization and we tried MLPs with 3 and 4 hidden layers. We selected the MLP with 4 layers since it performed better. The regularization constants for LR

Table 1: Comparison of all methods for Charades dataset

| | mAP | LRAP | JI |
|---|---|---|---|
| **CNN** (Feichtenhofer et al., 2019) | 0.3884 | 0.5347 | 0.2937 |
| **DRF - GS** | 0.2654 | 0.4386 | 0.2244 |
| **DRF - ILP** | 0.1926 | 0.2782 | 0.3063 |
| **DRF - IJGP** | 0.3115 | 0.4370 | 0.3194 |
| **DDN - LR - Pipeline** | 0.3450 | 0.4837 | 0.2897 |
| **DDN - LR - Joint** | 0.3902 | 0.5479 | 0.3025 |
| **DDN - MLP - Pipeline** | 0.3755 | 0.5493 | 0.2950 |
| **DDN - MLP - Joint** | **0.4052** | **0.5696** | **0.3356** |

Table 2: Comparison of all methods for TACoS and Wetlab datasets

| | TACoS | | | | Wetlab | | | |
|---|---|---|---|---|---|---|---|---|
| | mAP | LRAP | SA | JI | mAP | LRAP | SA | JI |
| **CNN** (Szegedy et al., 2016) | 0.7012 | 0.8076 | 0.4017 | 0.6079 | 0.7911 | 0.8208 | 0.3526 | 0.6383 |
| **DRF - GS** | 0.5584 | 0.7940 | 0.4693 | 0.6499 | 0.5392 | 0.7567 | 0.3532 | 0.5154 |
| **DRF - ILP** | 0.4026 | 0.6720 | 0.5094 | 0.6471 | 0.6304 | 0.7339 | 0.5967 | 0.7271 |
| **DRF - IJGP** | 0.5609 | 0.8141 | 0.4387 | 0.7010 | 0.7875 | 0.8525 | 0.5801 | 0.7366 |
| **DDN - LR - Pipeline** | 0.7164 | 0.8264 | 0.5042 | 0.6717 | 0.775 | 0.8549 | 0.5734 | 0.7018 |
| **DDN - LR - Joint** | 0.7463 | 0.8390 | 0.5371 | 0.6861 | 0.8432 | 0.869 | 0.6343 | 0.779 |
| **DDN - MLP - Pipeline** | 0.7294 | 0.8602 | 0.5785 | 0.6948 | 0.8116 | 0.8819 | 0.6184 | 0.7267 |
| **DDN - MLP - Joint** | **0.7797** | **0.8752** | **0.5963** | **0.7040** | **0.8809** | **0.8969** | **0.697** | **0.7925** |

(chosen from the set $\{0.1, 0.01, 0.001\}$) and the number of layers for MLPs were chosen via cross-validation. For joint learning, we chose the value of $\lambda$ (see equation 3) via cross-validation and reduced the learning rates of both LR and MLP models by expanding on the learning rate scheduler given in PySlowFast (Fan et al., 2020). We used the steps with relative learning rate policy, where the learning rate depended on the current epoch, and was chosen in the range of $[1e-3, 1e-5]$. We have provided the hyper-parameters used to do the experimentation in the configuration files (which is provided with the supplementary material).

## 4.3 RESULTS

We compare the baseline CNN with three versions of DRFs and four versions of DDNs using the four metrics and three datasets given in section 4.1. The results are presented in tables 1 and 2.

**Comparison between Baseline CNN and DRFs.** We observe that IJGP and ILP outperform the baseline CNN on the two non-probabilistic metrics JI and SA. IJGP typically outperforms GS and ILP on JI while ILP, because it performs accurate maximum-a-posteriori inference, outperforms IJGP and GS on SA (notice that SA is 1 if there an exact match between predicted and true labels and 0 otherwise and thus accurate MAP inference on an accurate model is likely to yield high SA). However, on metrics which require estimating the posterior probabilities, mAP and IRAP, the DRF schemes sometimes hurt and at other times are marginally better than the baseline CNN. We observe that advanced inference schemes, particularly IJGP and ILP are superior on average to GS.

**Comparison between Baseline CNN and DDNs.** We observe that the best performing DDN model, DDN-MLP with joint learning outperforms the baseline CNN on all the three datasets and on all metrics. Sometimes the improvement is substantial (e.g., 9% improvment in mAP on the wetlab dataset). Roughly speaking, the MLP versions are superior to the LR versions, and the joint models are superior to the ones trained using the pipeline method (in some cases because of joint learning the LR models outperform the pipeline MLP models). On the non-probabilistic metrics, the pipeline models are often superior to the baseline CNN while on the mAP metric they may hurt the performance.

**Comparison between DRFs and DDNs.** We observe that the jointly trained DDNs are almost always superior to the best performing DRFs on all datasets. Interestingly, on average, the pipeline DDN models outperform the DRF models when performance is measured using the mAP and LRAP

metrics. However, when the SA and JI metrics are used, we observe that there is no significant difference in performance between pipeline DDNs and DRFs.

In summary, jointly trained deep dependency networks are superior to the baseline neural networks as well as networks that combine Markov random fields and neural networks. This clearly demonstrates the practical usefulness of our proposed method.

## 5 RELATED WORK

A large number of methods have been proposed that train PGMs and NNs jointly. For example, Zheng et al. (2015) proposed to combine conditional random fields (CRFs) and recurrent neural networks (RNNs), Schwing & Urtasun (2015); Larsson et al. (2017; 2018); Arnab et al. (2016) showed how to combine CNNs and CRFs, Chen et al. (2015) proposed to use densely connected graphical models with CNNs, and Johnson et al. (2016) combined latent graphical models with neural networks. As far as we know, ours is the first work that shows how to jointly train a dependency network, neural network hybrid. Another virtue of DDNs is that they are easy to train and parallelize, making them an attractive choice.

The combination of PGMs and NNs has been applied to improve performance on a wide variety of real-world tasks. Notable examples include Semantic Segmentation (Arnab et al., 2018; Guo & Dou, 2021), human pose estimation (Tompson et al., 2014; Liang et al., 2018; Song et al., 2017; Yang et al., 2016), semantic labeling of body parts (Kirillov et al., 2016), stereo estimation Knöbelreiter et al. (2017), language understanding (Yao et al., 2014), joint intent detection and slot filling (Xu & Sarikaya, 2013), 2D Hand-pose Estimation (Kong et al., 2019), depth estimation from a single monocular image (Liu et al., 2015), Polyphonic Piano Music Transcription (Sigtia et al., 2016), face sketch synthesis Zhu et al. (2021), Sea Ice Floe Segmentation (Nagi et al., 2021) and Crowdsourcing aggregation (Li et al., 2021)). As far as we know, ours is the first work that uses jointly trained PGM+NN combinations to solve the multi-label action classification task in videos.

To date, dependency networks have been used to solve various tasks such as collective classification (Neville & Jensen, 2003), binary classification (Gámez et al., 2006; 2008), multi-label classification (Guo & Gu, 2011), part-of-speech tagging (Tarantola & Blanc, 2002), relation prediction (Figueiredo et al., 2021) and collaborative filtering (Heckerman et al., 2000). Ours is the first work that combines DNs with sophisticated feature representations and performs joint training over these representations.

## 6 CONCLUSION AND FUTURE WORK

More and more state-of-the-art methods for challenging applications of computer vision tasks usually use deep neural networks. Deep neural networks are good at extracting features in vision tasks like image classification, object detection, image segmentation, and others. Nevertheless, for more complex tasks involving multi-label classification, these methods cannot model crucial information like inter-label dependencies. In this paper, we proposed a new modeling framework called deep dependency networks (DDNs) that combines a dependency network with a neural network and demonstrated via experiments on three video datasets that it outperforms the baseline neural network, sometimes by a substantial margin. The key advantage of DDNs is that they explicitly model and reason about the relationship between the labels, and often improve model performance without considerable overhead. In particular, DDNs are simple to use, admit fast learning and inference, are easy to parallelize, and can leverage modern GPU architectures.

Avenues for future work include: applying the setup described in the paper to multi-label classification, new tasks in computer vision, natural language understanding and speech recognition; developing advanced inference schemes for dependency networks; converting deep dependency networks to Markov random fields for better inference (cf. (Lowd, 2012)); etc.

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

## A APPENDIX

### A.1 COMPARING TRAINING AND INFERENCE TIME

Table 3: Time Comparisons for the Proposed methods. Learning Time is in Hours and Inference times is in seconds. Inference was performed on a CPU while training was performed on a GPU.

|  | Tacos | | Wetlab | | Charades | |
|---|---|---|---|---|---|---|
|  | **Train** | **Inference** | **Train** | **Inference** | **Train** | **Inference** |
| **DRF - GS** | ∼ 5 hrs | ∼ 0.58 sec | ∼ 6.5 hrs | ∼ 0.61 sec | ∼ 8 hrs | ∼ 1.93 sec |
| **DRF - ILP** | ∼ 5 hrs | ∼ 1.54 sec | ∼ 6.5 hrs | ∼ 1.46 sec | ∼ 9 hrs | ∼ 2.42 sec |
| **DRF - IJGP** | ∼ 5 hrs | ∼ 2.31 sec | ∼ 6.5 hrs | ∼ 2.15 sec | ∼ 9 hrs | ∼ 5.8 sec |
| **DDN - LR - Pipeline** | ∼ 1.5 hrs | ∼ 0.1 sec | ∼ 2 hrs | ∼ 0.15 sec | ∼ 3 hrs | ∼ 0.39 sec |
| **DDN - LR - Joint** | ∼ 6 hrs | ∼ 0.1 sec | ∼ 7.25 hrs | ∼ 0.15 sec | ∼ 12 hrs | ∼ 0.39 sec |
| **DDN - MLP - Pipeline** | ∼ 2.25 hrs | ∼ 0.19 sec | ∼ 3 hrs | ∼ 0.31 sec | ∼ 4.25 hrs | ∼ 0.58 sec |
| **DDN - MLP - Joint** | ∼ 7 hrs | ∼ 0.19 sec | ∼ 8 hrs | ∼ 0.31 sec | ∼ 14.5 hrs | ∼ 0.58 sec |

Let us compare different methods based on the information given in table 3.

### A.1.1 COMPARISON AMONG DRFs

The learning time for the DRFs remains the same across the different methods, because we use the same models and apply different inference techniques on them. But for inference time, we can see that as we use more sophisticated methods, the inference time goes up.

### A.1.2 COMPARISON AMONG DDNs

For DDNs the inference times remain the same for both, pipeline and joint models. But for learning we see that joint model takes more time than the pipeline model. This is due to the fact that we are jointly learning the NN and the DN, and the inclusion of NNs for learning drives the time up.

### A.1.3 COMPARISON BETWEEN DRFs AND DDNs

The learning time for pipeline DDN models are significantly less than that of the DRF model. Both the pipeline model and the joint model are exceptionally faster than the DRF methods. These two observations confirm our comments that DDNs are very fast and thus can be used in real time.

