# OpenReview forum: "Deep Dependency Networks for Action Classification in Video"
_ICLR.cc/2023/Conference — Submitted to ICLR 2023_

### Official Review · Reviewer_KzJi · 2022-10-24

**Confidence:** 4
**Correctness:** 3
**Technical Novelty And Significance:** 2
**Empirical Novelty And Significance:** 2
**Recommendation:** 3

**Clarity, Quality, Novelty And Reproducibility:**

Clarity: poor; the paper is unclear and hard to understand
Quality: poor,
Novelty: poor; the paper proposes a simple dependency network with a joint training method, which does not show significant novelty
Reproducibility: poor; the paper lack details about their design, and it is hard to reproduce if the authors do not release the code


**Strength And Weaknesses:**

Strength
+ Well motivated in Introduction

Weaknesses
- Unclear diagram, it is hard to figure out the joint learning methodology and how the LOSS is designed according to the figure.
- Hard to understand due to typo/unclaimed terminology (e.g., 'ds outperforms CNN' in Preliminaries). Meanwhile, the advantages claimed in this work are mainly explained on conceptual/abstract induction or assumption instead of convincing analysis (e.g., 'An issue with the loss function just described is that it significantly changes the structure and semantics of the output nodes of the neural network').
- Experiments lack a detailed description of author-design modules (even with an unclear setting like 'MLPs with 3 or 4 hidden layers), which makes the experimental procedure unconvincing.
- For some parts of the experiment results, baseline like Slowfast seems to have higher performance in public disclosure compared with the effect provided here, which is confusing.
- The authors state the work is a method for action classification. However, the majority of the work lies on optimizing the correlation of feature-label and label-label, while the adjustment for the action classification task is not significant; in other words, I prefer to categorize the work as an application of the previous general method other than insightful research in the scope of action recognition. Therefore, the novelty is relatively limited, in my opinion. Also, if the work is mainly effective in solving the 'MLAC problem, the title of the paper is too big for the work done.


**Summary Of The Paper:**

This paper proposes a simple joint training deep dependency network, which can be built upon CNN baseline models to improve the performance in multi-label action classification tasks. The authors conduct experiments on three open datasets (Charades, TACoS, and Wetlab) and show their superiority on four metrics (mAP, LRAP, SA, and JI)

**Summary Of The Review:**

The work lack novelty and convincing experiment detail, also the paper is hard to understand and the illustration is full of assumption and conceptual description.

---

> ### Author Response · Authors · 2022-11-18
> **Responses to Reviewer KzJi**
>
> We thank the reviewer for the critical feedback and valuable suggestions. We address the specific questions in two parts.
>
> ### Question 1
> Unclear diagram, it is hard to figure out the joint learning methodology and how the LOSS is designed according to the figure.
> ### Reply
> In figure 2 -
> $E_i$  are the output nodes of the neural network of the DDN. While $X_i$ are the class label nodes for the Dependency Network. As mentioned in section 3.1, each $X_i$ is conditionally dependent on $X_{-i}, E$. Thus the computational graph provides us with the way in which we will compute the values at the $X_i$ nodes (forward pass), and then we can apply back-propagation on this computational graph to calculate the gradient with respect to all weights of the DDN and the parameters of the CNN.
>
> The loss $L_{i,DDN}$ (given in equation 2) is applied to each node of the Dependency Network. It is used to update the $\theta_{ij}$ and $\pi_{ij}$ weights of the Dependency Network, and also the gradients are back-propagated through the neural network as well (since $E_i$ are also input to our dependency networks).
> $$ L_{i,DDN}= - x_i \log P_i(X_i=1|e,x_{-i}; \Theta_i,\Pi_i) - (1-x_i) \log P_i(X_i=0|e,x_{-i}; \Theta_i,\Pi_i)  $$
>
> The loss $L_{i,NN}$ (equation 3 in the paper) is the loss at the $i^{th}$ output node of the CNN. Thus it depicts the loss at each neural network output node. The first part of the loss corresponds to the Binary Cross Entropy loss that is applied to the outputs from the CNN (which was also used to train the baseline CNN models) and the second part of the loss comes from the Dependency Networks ($L_{1,DNN}$). Since there is a connection going from each $E_i$ to all of the $X_i$ nodes, we have done the sum in the second part of the equation ($\frac{\lambda}{n} \sum_{j=1}^{n} \mathcal{L}_{j,DDN}  $).
>
> $$ L_{i,NN}= ( - x_i \log e_i - (1-x_i) \log (1-e_i)) + \frac{\lambda}{n} \sum_{j=1}^{n} \mathcal{L}_{j,DDN}  $$
>
> In the figure $E_1, E_2$ and $E_3$ are the output nodes of the feature extractor. Thus if we take the node $E_1$, then we would backpropagate the gradients that come from the loss $L_{1,NN}$. While $X_1, X_2$ and $X_3$ are the class label nodes in the Dependency Network.  Thus if we take the node $X_1$, then we would backpropagate the gradients that come from the loss $L_{1,DNN}$ on the network that represents the probability distribution for $X_1$.
>
>
>
> ### Question 2
> Hard to understand due to typo/unclaimed terminology (e.g., 'ds outperforms CNN' in Preliminaries). Meanwhile, the advantages claimed in this work are mainly explained on conceptual/abstract induction or assumption instead of convincing analysis (e.g., 'An issue with the loss function just described is that it significantly changes the structure and semantics of the output nodes of the neural network').
> ### Reply
> We have removed the comment - 'ds outperforms CNN' from Preliminaries. Regarding conceptual/abstract induction, let us clarify. We can view each i-th output node of a neural network as representing $P(E_i|v)$ where $v$ is a video segment, and each output node of the DDN as representing $P(X_i|e_i,x_{-i},v)$.  Since we are using a pre-trained model, we do not want large parameter gradients to rapidly undo the current solution. Thank you for pointing it out. We will rephrase it. Equation 3 accomplishes the objective mentioned above.
>
> ### Question 3
> Experiments lack a detailed description of author-design modules (even with an unclear setting like 'MLPs with 3 or 4 hidden layers), which makes the experimental procedure unconvincing.
> ### Reply
> By the line 'MLPs with 3 or 4 hidden layers', we mean that the number of hidden layers was a hyper-parameter of the model and we tried the values mentioned above and selected the one which gave the best score. As mentioned in the title of that part, we have provided all the hyper-parameters (in config files) that we chose to get the best model.
>
> ### Question 4
> For some parts of the experiment results, baseline like Slowfast seems to have higher performance in public disclosure compared with the effect provided here, which is confusing.
> ### Reply
> The performance mentioned in the paper is the same as the one mentioned on the [py-slowfast repository](https://github.com/facebookresearch/SlowFast/blob/main/MODEL_ZOO.md#charades). The reason behind the difference between the metrics reported in public disclosure and the metrics reported in the paper is that they are reported on different splits of the data. In the public disclosure, evaluation was reported on the test set. However, the metrics reported in the pyslowfast repository and this paper are evaluated on the validation dataset.

---

> > ### Author Response · Authors · 2022-11-18
> > **Responses to Reviewer KzJi - Part 2**
> >
> > ### Question 5
> > The authors state the work is a method for action classification. However, the majority of the work lies on optimizing the correlation of feature-label and label-label, while the adjustment for the action classification task is not significant; in other words, I prefer to categorize the work as an application of the previous general method other than insightful research in the scope of action recognition. Therefore, the novelty is relatively limited, in my opinion. Also, if the work is mainly effective in solving the 'MLAC problem, the title of the paper is too big for the work done.
> > ### Reply
> > You are right that we have a general method but we have only applied it to MLAC. As far as we are aware, deep dependency networks have not been proposed or evaluated in literature. We can apply it to multi-label classification or in general to model $$P(x_1,\ldots,x_n|v)$$
> >
> > One of the concerns that the reviewer KzJi mentioned was that the reproducability was poor and that it would be hard to reproduce if the authors do not release the code.  But we have provided the code and configuration files to replicate the results shown in the paper as supplementary material.
> >
> > We appreciate your critical review. It will help us improve the paper. We disagree with the novelty part but we feel that we have addressed other issues in this response.

---

### Official Review · Reviewer_apzy · 2022-10-25

**Confidence:** 4
**Correctness:** 3
**Technical Novelty And Significance:** 2
**Empirical Novelty And Significance:** 2
**Recommendation:** 5

**Clarity, Quality, Novelty And Reproducibility:**

Strength
 + Overall clear presentation of motivation, design, experiments, and results

Weakness
  + In the best performing setting (`DDN-MLP-Joint`), is the network architecture the same between `CNN` and `DDN-MLP-Joint`? Basically a CNN for extracting features and a MLP for mapping features to multi-class predictions. What's different between the two is only the losses applied, is this understanding correct?
  + need to remove a comment at the beginning of Section 2: "So just a quick question ......"

**Strength And Weaknesses:**

Strength
 + This work noticed a particularly interesting problem on the simple implementation of `CNN + MRF + GS/ILP/IJGP` for multi-label action classification, i.e., they yield poor posterior probability estimates.
 + The proposed approach surely addresses the discovered problem and outperforms the aforementioned baseline.

Weakness
 + the baselines used in experiments are out of date. On Charades dataset, the CNN baseline is from 2019. On TACoS and Wetlab, the baseline is from 2016. While the experiments w.r.t. these baselines can provide some proof of usefulness of the proposed method. It is unclear if this performance gain can hold against more recent deep learning models.


**Summary Of The Paper:**

This work experiments with hybrid modeling with CNN and PGM for the task of multi-label action classification.
This work considers (1) `CNN` and (2) `CNN + MRF + GS/ILP/IJGP` as two baselines.
This work proposes an approach called `Deep Dependency Networks` to yield better posterior probability estimates.
Experiments show the proposed approach output performs baseline (1) and (2) in terms multi-label action classification metrics.

**Summary Of The Review:**

Overall I like the presentation of the problem motivation and the proposed solution. However the baselines used in experiments are not up to date. It's unclear whether the performance gain hold against more recent deep learning models.

---

> ### Author Response · Authors · 2022-11-18
> **Responses to Reviewer apzy**
>
> We thank the reviewer for the insightful review and bringing about interesting points of discussion. We answer each of the required clarifications below.
>
> ### Question 1
> The baselines used in experiments are out of date. On Charades dataset, the CNN baseline is from 2019. On TACoS and Wetlab, the baseline is from 2016. While the experiments w.r.t. these baselines can provide some proof of usefulness of the proposed method. It is unclear if this performance gain can hold against more recent deep learning models.
> ### Reply
> We chose these baselines because we wanted to show that DDNs can work with two different kinds of feature extractors. For TACoS and Wetlab datasets, we did the experiments using 2D CNNs because the datasets are not large, and thus 3D CNNs are most likely to overfit the training dataset. Therefore we chose one of the SOTA 2D CNNs for image classification and used it as a baseline. The reason for choosing these datasets was that one was from the cooking domain, and one had laboratory experiments. This helps us to show that the model does not depend on the domain of the Dataset and can work with various kinds of settings and environments.
>
> For the Charades dataset, most of the state-of-the-art methods did not provide *official training scripts* or trained models (usually, you would find models that are trained on the Kinetics-400, SSv2, or ImageNet-1K datasets in the repositories, but not on the Charades Dataset). However, the official repository of the SlowFast networks (PySlowFast) provides the trained model and the training and testing code for the Dataset. This helps us find whether DDNs is the reason behind the improvement! Again, look at DDNs as a light-weight method (computationally speaking) that can yield improvements over CNNs/neural networks.
>
> We wanted the official training and testing code because we wanted to do Joint Learning, for which we need to have the correct pre-trained models and training scripts. We wanted to do joint training in exactly the same enviroment as the models were trained before; otherwise, we would not have known if the improvements are due to DDNs, the differences in the training scripts, or the hyper-parameters.
>
> Most of the current deep learning models do not consider the inter-label dependencies for multi-label classification. They only model the input-label relationships. Moreover, to even show that DDNs work on baselines that take different kinds of inputs, we did experimentation with 2D CNNs and 3D CNNs. The main reason to use DDNs on top of these deep learning models is to improve their performance on the multi-label activity classification task. The reason behind the improvement is that they model the inter-label dependencies. And the DDN model can be used on top of any other feature extractor, and since it models the inter-label dependencies, it is most likely to improve on the performance of the baseline.
>
>
> ### Question 2
> In the best performing setting (DDN-MLP-Joint), is the network architecture the same between CNN and DDN-MLP-Joint? Basically a CNN for extracting features and a MLP for mapping features to multi-class predictions. What's different between the two is only the losses applied, is this understanding correct?
> ### Reply
> No, for `DDN-MLP-Joint`, we have a CNN as the feature extractor, and then we have **dependency network** which maps these features to *multi-label predictions*. As mentioned in the Preliminary Section, a dependency network represents the joint distribution using a set of conditional probability distributions, one for each variable. Each conditional distribution defines the probability of a variable given all of the others.
> $$ p\left(x_i \mid Pa_G(x_i)\right)=p\left(x_i \mid x_{-i} \right)$$
>
> Thus, we will have |labels| MLP networks, one for each label and not just a single model. Since we are using Dependency Networks, we cannot just do a forward pass on the MLPs and get the outputs; we need to use Gibbs sampling for inference (because we have a probability distribution and are trying to find the max-marginal). Dependency Networks are a type of probabilistic graphical model which helps us to model the relationships between the labels (which is not done at the feature extractor level)
>
> It is the same for the `DDN-LR` models as well. We are not using just an LR model to process the features extracted by the baseline but applying a probabilistic graphical model on top of it (which helps to model the relationships between labels) to get the output labels.
>
> ### Question 3
> need to remove a comment at the beginning of Section 2: "So just a quick question ......"
> ### Reply
> Thank you for pointing this out. We have removed the comment.

---

### Official Review · Reviewer_rXD3 · 2022-10-25

**Confidence:** 3
**Clarity, Quality, Novelty And Reproducibility:** I am satisfied with the clarity, qual…
**Correctness:** 3
**Technical Novelty And Significance:** 3
**Empirical Novelty And Significance:** 3
**Recommendation:** 6

**Strength And Weaknesses:**

Strength:
- DDNs are simple and easy to use, which is GPU friendly.

- DDNs can model and reason about the relationship between the labels.

Weakness:
- The transformer model can also be regarded as modeling the graph between tokens. I suggest the author should compare with transformer-based models.

**Summary Of The Paper:**

This paper proposes a simple approach that combines the strengths of probabilistic graphical models and deep learning architectures for solving the multi-label video classification task. This work proves that jointly learning the proposed model can yield significant improvements in performance over the baseline neural network. They do experiments on three video datasets: Charades, Textually Annotated Cooking Scenes (TACoS), and Wetlab, showing that deep dependency networks are almost always superior to pure neural architectures that do not use dependency networks.


**Summary Of The Review:**

This paper proposes a new hybrid model called DDNs, combining the strengths of dependency networks.

---

> ### Author Response · Authors · 2022-11-18
> **Responses to Reviewer rXD3**
>
> Thank you for the suggestions. We address specific questions below.
> ### Question 1
> The transformer model can also be regarded as modeling the graph between tokens. I suggest the author should compare with transformer-based models.
> ### Reply
> We did not use the transformer model because we cannot exploit the parallelizable advantages of the dependency networks if there is a single model that is used to create the label-label dependencies. We wanted the DDNs to add little to no overhead over the baseline. As a result of this, we could get significant improvements and not have to think about the computational requirements.
>
> The baselines (especially Slowfast) takes a lot of time to train. Transformer models would incur a considerable overhead on top of these models, specifically while jointly learning them with CNN.
>
> A key advantage of our approach is that it is light-weight and thus does not require significant computation power. All of our experiments were performed on a machine with just one A40 GPU. The SlowFast network requires 40GBs of the A40 and thus we wanted our method to not add a huge overhead.
>
> Thank you again for your suggestion; we plan to combine transformer models with dependency networks in order to model both inter-label and label-feature relationships in future work.

---

### Decision · Program_Chairs · 2023-01-20

**Decision:**

Reject

**Justification For Why Not Higher Score:**

- Several critical issues in the empirical evaluation
- Clarity, limited scope and novelty
- Worries about reproducibility given the lack of open-sourced code and other details

**Justification For Why Not Lower Score:**

N/A

**Metareview: Summary, Strengths And Weaknesses:**

The authors propose a novel approach which jointly trains a conditional dependency network and a deep neural network for multi-label action recognition in video. The idea is to use the DNN for feature extraction and the dependency network represents to reasons about the probabilistic relationships between the labels and the extracted features. Empirical evaluation is performed on Charades, Textually Annotated Cooking Scenes (TACoS), and Wetlab and show that the proposed method improves the results across multiple benchmarks.

The reviewers appreciated the motivation behind the problem, but raised several critical concerns. (1) Empirical evaluation against transformer-based models which essentially model the dependencies between tokens with an implicit graph. (2) Empirical evaluation with (severely) out-of-date baselines. (3) Clarity of writing, undefined terminology. (4) Limited scope and novelty in the current instantiation. (5) Worries about reproducibility given the lack of open-sourced code and other details. These concerns were not satisfyingly addressed in the rebuttal and I will hence recommend rejection.